# Development of a Novel Tape-Casting Multi-Slurry 3D Printing Technology to Fabricate the Ceramic/Metal Part

**DOI:** 10.3390/ma16020585

**Published:** 2023-01-06

**Authors:** Cho-Pei Jiang, Yulius Shan Romario, Ehsan Toyserkani

**Affiliations:** 1Department of Mechanical Engineering, National Taipei University of Technology, Taipei 10608, Taiwan; 2Graduate Institute of Manufacturing Technology, National Taipei University of Technology, Taipei 10608, Taiwan; 3Department of Mechanical and Mechatronics Engineering, University of Waterloo, Waterloo, ON N2L 3G1, Canada

**Keywords:** additive manufacturing, multi-material, tape casting, 3D printing

## Abstract

Printing ceramic/metal parts increases the number of applications in additive manufacturing technology, but printing different materials on the same object with different mechanical properties will increase the difficulty of printing. Multi-material additive manufacturing technology is a solution. This study develops a novel tape-casting 3D printing technology that uses bottom-up photopolymerization to fabricate the green body for low-temperature co-fired ceramics (LTCC) that consist of ceramic and copper. The composition of ceramic and copper slurries is optimized to allow printing without delamination and sintering without cracks. Unlike traditional tape-casting processing, the proposed method deposits two slurries on demand on a transparent film, scrapes it flat, then photopolymerization is induced using a liquid crystal displayer to project the layer pattern beneath the film. The experimental results show that both slurries have good bonding strength, with a weight ratio of powder to resin of 70:30, and print a U-shaped copper volume as a circuit within the LTCC green body. A three-stage sintering parameter is derived using thermogravimetric analysis to ensure good mechanical properties for the sintered part. The SEM images show that the ceramic/copper interface of the LTCC sintered part is well-bonded. The average hardness and flexural strength of the sintered ceramic are 537.1 HV and 126.61 MPa, respectively. Volume shrinkage for the LTCC slurry is 67.97%, which is comparable to the value for a copper slurry of 68.85%. The electrical resistance of the printed copper circuit is 0.175 Ω, which is slightly greater than the theoretical value, hence it has good electrical conductivity. The proposed tape-casting 3D printer is used to print an LTCC benchmark. The sintered benchmark part is validated for the application in the LTCC application.

## 1. Introduction

Additive Manufacturing (AM), which is also known as three-dimensional (3D) printing, is a method to fabricate physical parts from a virtual design model. The American Society for Testing and Materials (ASTM) classifies AM into seven categories. The materials for AM are polymers, ceramics, and metals in an initial form of liquid, powder, or filament. Most printers use a single material to build the printed model, but multi-material printing is used for functional structures and to create visually appealing prototypes [1]. 

In terms of ceramic components, multi-nozzle fused deposition modeling (FDM) is used to extrude the material to allow multi-ceramic printing [2]. Filaments with a lower content of ceramic powder are frequently used to allow the smooth passage of the fuse material through the nozzle and to increase the bond strength between the layers. Ceramic green parts are printed, but there is significant volume shrinkage during sintering, thus cracks and severe deformations are possible. Vat polymerization (VP) is used to print multi-material objects [3]. VP methods and photocurable slurries that use different materials are being continuously developed.

Multi-material 3D printing using VP allows objects to be printed using different resins or slurries. A multi-vat 3D printer using the bottom-up method has been developed to print Tai-Chi models, using two resins with the same composition and differently colored dyes [4]. There is no delamination because the shrinkage ratio for these two resins is the same. Inter-staining of the interface between these two cured resins is avoided by using a cleaning module to remove the residual resin from the printed object [5]. A complete denture can be printed using multi-vat 3D printing because the process uses two resins with different compositions and colors to construct an object [6]. Most materials used for a 3D printer for bottom-up VP are photocurable resins, but few photocurable slurries are available because the slurries have a suspension problem. 

The slurry is mainly composed of a photocurable resin and powder, but the suspension of the powder in the slurry determines whether an object can be printed. Aluminum oxide (Al_2_O_3_) has a density of 3.95 g/cm^3^; therefore, ceramic powder is used as a slurry because it has a high temperature and is resistant to corrosion [7,8]. Zirconia oxide (ZrO_2_) has a density of 5.68 g/cm^3^ and is used for printing; however, its high density results in poor suspension, making it difficult to use for a bottom-up VP-type 3D printer [9,10]. 

An object printed successfully is called a green body, which is debinded by thermal sintering to produce a highly dense object. Previous studies have shown that the density, size, and content of powder particles in the slurry will affect the viscosity, fluidity, and suspension of the particles in the slurry. Poor suspension indicates particles in the slurry sink to the bottom of the vat quickly; thus, there is an inhomogeneous distribution in the printed object, liquid-solid separation during the printing process, and cracking during the sintering process [11]. 

For a bottom-up VP-type 3D printer, a slurry with high viscosity generates a high shearing force, and it causes defects on the corners of objects occur when the platform is raised from the slurry. Poor fluidity prevents the slurry from quickly backfilling the printing area, hence a scraper is required to stimulate the slurry to flow. A top-down VP-type 3D printer uses a slurry with a high-density powder to print a ceramic object [12]. 

A traditional top-down VP-type 3D printer prints multiple resins by using a rotating disk with multiple vats. Consequently, the residual resin must be cleaned from the printed object, and different resins must be separated to avoid contamination [3]. For multi-slurry printing, an effective cleaning method need to be done before the platform moves to another slurry vat to prevent contamination between the different slurry. The greater the volume of the printing object, the better the suspension of the slurry must be to avoid inhomogeneous particle distribution in the printed object. Generally speaking, the viscosity coefficient of the slurry is higher than the original resin without adding powder. Therefore, if the object has small or deep holes, the slurry filled in the holes is not easy to flow out and remove during the cleaning process. These two methods for multi-slurry printing require a low-viscosity resin and adequate suspension of the slurry. In addition, contamination must be avoided when changing the slurry to build the object. 

Multi-slurry 3D printing processes decrease the fabrication time and allow a more flexible manufacturing process than conventional fabrication methods for Low-Temperature Co-fired Ceramics (LTCC). An LTCC is usually fabricated by producing ceramic and conductor slurries to cast a 50 µm thickness layer on the tape. The casting tape is dried and rolled as two rolls of tape. The ceramic tape is punched to the required dimensions to create the vacant portion, and the shape is selectively blanked using the conductor tape to fill the vacant portion. The punched ceramic tape and blanked conductor tape are assembled to form a layer. This process is repeated until all layers are staked to construct the green body for the LTCC. The LTCC is produced by co-firing the green body. Discarded ceramic and conductor tape are waste materials and cannot be recycled. This study proposes a novel tape-casting multi-slurry 3D printing technology to print a ceramic/metal part for an LTCC application using no drying or rolling process. The slurry can also be recycled and reused.

This study develops a novel tape-casting multi-slurry 3D printer and optimizes the composition of two photocurable slurries to print the green body for an LTCC with good bonding strength. The effect of varying particle weight percentages in the slurry on the success rate for printing and the mechanical properties of the printed green body are analyzed. A three-stage sintering parameter is proposed, and the microstructure of the ceramic/copper interface of sintered part is observed.

## 2. Materials and Methods

### 2.1. Principle of Tape-Casting Multi-Slurry 3D Printing Technology

Unlike a traditional tape-casting process that requires two rolls of tape for ceramic/metal object fabrication, the proposed method uses one transparent tape. It deposits the photocurable slurry on it on demand. Figure 1a shows a liquid crystal display (LCD) with a resolution of 2160 × 3840 (LS055D1SX05, SHARP Co., Osaka, Japan) and a display dimension of 68.04 × 120.96 (mm^2^).The light emitting diode (LED) module has a 10 Watt output power with a 405 nm wavelength. Both are assembled as the layer pattern generator and set up beneath the highly transparent tape. The layer pattern exposes the slurry on the tape to induce photopolymerization. 

A platform on a driving screw powered by a motor rotates 90 degrees around the *Z*-axis when cleaning is required. Two scrapers are used to separate scrapping the ceramic and copper slurries to prevent the slurries from mixing. Figure 1b shows the design for the tape-casting multi-slurry 3D printer. Two containers with blenders filled with ceramic and copper slurries deposit slurry on the tape on demand. After the material is deposited on the tape, the tape will move until a new print area is right above the LCD, then the LCD will display the pattern generated by the layer pattern generator. This process will repeat layer by layer until it finishes printing.

### 2.2. Slurry Preparation

The proposed process is used to produce an LTCC, which consists of LTCC ceramic and copper materials. The photocurable slurry is mainly composed of photocurable resin and powder, thus this study optimizes the ceramic and copper slurries by measuring the success rate for printing using different weight percentages of powder and photocurable resin. 

HDDA (1,6-Hexanediol Diacrylate) and TMPTA (Trimethylolpropane triacrylate) are frequently used for VP-type 3D printers [4,10,13]. The viscosity of HDDA is less than that of TMPTA, but the cured strength of TMPTA is greater than that of HDDA. Therefore, this study uses a weight ratio of HDDA (Double Bond Chemical Co., New Taipei, Taiwan) and TMPTA (Double Bond Chemical Co., New Taipei City, Taiwan) of 1:2 for the resin material, and a photoinitiator (Irgacure 819, ACT Chemical Co., New Taipei, Taiwan) is added to produce a photocurable resin. 

To produce the slurry, the weight ratios of photocurable resin and powder are 50:50, 60:40, 70:30, and 80:20. The powders are LTCC ceramic powder with a grain size of less than 2 µm (CCF8-LTCC, Ferro Co., Taipei, Taiwan) and copper powder with an average particle size of 20 µm (Emperor Chemical Co., Taipei, Taiwan). The mixed photocurable slurries are ball milled in a zirconia jar using zirconia balls at 150 rpm for 6 h to produce a highly homogeneous and refined powder.

### 2.3. Curing Test

The curing test optimizes the exposure time to ensure an appropriate curing depth because the color and particle size of both slurries are different. During the curing test, the synthesized slurries are deposited on the transparent film and exposed to a 50 × 50 mm^2^ pattern using the printer for different periods. The cured objects are mounted in an epoxy mold and then cut in half to measure the curing depth.

### 2.4. Microstructural Observation

A field emission scanning electron microscope (FE-SEM JSM-7610F, Tokyo, Japan) is used to measure the powder size of the as-received ceramic and copper, the powder size after ball milling, and the microstructure of the fusion interface between the LTCC ceramic and copper layers is observed.

### 2.5. Thermogravimetric Test and Sintering Treatment

A two-stage sintering process is frequently used for a 3D-printed green body: the first stage decomposes the cured resin and the second stage fuses the particles [10,14]. To optimize the sintering parameters, a thermogravimetric test (TGA) is used to plot the weight against the temperature of the slurry. This curve is used to calculate the temperature at which resin evaporates, but it is necessary to ensure that no bubbles are generated during the first stage of sintering by controlling the heating rate to allow the grains to grow without cracks and increase the strength after the second stage of sintering. Therefore, the temperature and heating rate of the first stage must be sufficient to evaporate the cured resin, but the evaporation rate must not allow bubbles to be generated. The temperature of the second stage is increased and maintained for an appropriate time at the temperature at which the ceramic and copper powders melt and crystallize. 

In this study, thermogravimetric analysis of the slurry will be performed to optimize sintering parameters. As the current two-stage sintering method is only used for a single material, it is unknown whether it is suitable for co-sintering ceramic/metal materials. Therefore, sintering parameters suitable for ceramic/metal co-sintering will be proposed.

### 2.6. Shrinkage Analysis and Density Measurement

The printed green body mainly contains resin and powder, but the resin is vaporized during the first stage of sintering. The powder is fused at the second stage during sintering when the temperature increases, resulting in crystallization. The density then becomes greater than that of the printed green body, and the volume shrinks. This study uses two slurries: metal and ceramic. 

The density of these two materials differs significantly, such that the interface can feature delamination due to the difference in the shrinkage ratios after sintering. Therefore, it is necessary to measure the difference in the shrinkage ratio along each axis before and after the printed green bodies are sintered using two different slurries to determine the difference in the deformation of printed parts after sintering. This study prints five cubic samples using these two slurries. The size of each sample is 10 mm × 10 mm × 5 mm. The shrinkage rate along each axis is calculated by measuring the dimensions of the printed green body and the sintered part. Calipers are used to measure the length along each axis.

In order to measure the density of the LTCC and copper sintered part, this study uses an electronic balance to determine the weight of the green body and the sintered part. The Archimedes principle, as shown in Equation (1) [10], is used to measure the density of sintered part using the results from an electronic densimeter (SD-200L, Alfa Mirage, Japan). The measurement resolution is 0.0001 g and the maximum capacity is 200 g. The respective theoretical densities of LTCC and copper are 5.1 and 8.94 g/cm^3^.
(1)$$density\;\left(D\right)=\frac{W_{air1}\times water\;density}{W_{air2\;}-W_{water}}$$
where $W_{air1}$ is the weight of the sintered part. The sintered part is soaked in water, the surface is wiped, the weight of the sintered part is measured again in the air, and this value is recorded as $W_{air2}$. The sintered part is then immersed in a tank of pure water with known weight, and the overall weight is calculated by subtracting the weight of the pure water tank to determine $W_{water}$.

### 2.7. Mechanical Property of the Sintered Part

LTCC is a brittle material, thus the flexural strength and hardness after sintering are measured using three-point bending (Universal Testing Machine CY-20, Chun-Yen Co., Taichung, Taiwan) and Vickers hardness (HMV-2000, Shimadzu, Kyoto, Japan) tests. Five pieces of the sintered specimen with dimensions of 10 mm × 10 mm × 15 mm are prepared. The test complies with the ASTM standard and is compared with the result of a previous study [15] to determine its applicability for an LTCC device. The average hardness is measured by indenting at least 5 points on each test piece. 

The three-point bending test uses five pieces of the sintered specimen with dimensions of 2 mm × 1.5 mm × 25 mm. The span length is 20 mm to comply with the ASTM standard. The pressure probe moves down at a speed of 0.1 mm/min, until the test bar breaks, and then the maximum load is measured by the probe and is used to calculate the flexural strength. Copper is frequently used for communication and electrical devices; thus, this study measures conductivity using a volt-ohm-milliammeter.

### 2.8. Benchmark

LTCC devices are fabricated using a multi-layer process, for which metal structures are printed onto the surface of green tapes, then multi-layered and densified at high temperatures (850–900 °C) to create ceramic components with electrical circuits. This study creates a copper tunnel as an electrical circuit in an LTCC device to determine the success rate for printing a green body, using the developed 3D printer with ceramic and copper slurries and the microstructure of the bonding interface, and the conductivity of the sintered part are determined. Figure 2 shows the proposed benchmark LTCC device with dimensions of 10 mm × 20 mm × 10 mm and the embedded tunnel with a diameter of 5 mm.

## 3. Results

### 3.1. Tape-Casting Multi-Slurry 3D Printer

Figure 3 shows the tape-casting multi-slurry 3D printing process for the assembled printer. Figure 3a shows the LTCC and copper slurries deposited on the tape, and the platform is moved down until the desired print layer thickness is achieved. Figure 3b shows the photopolymerized LTCC, which adheres to the platform. The photo on the right shows the mask for this print layer. When exposure is completed and the platform is raised, the photomask looks like the pattern of the printed LTCC slurry layer. Figure 3c shows the platform moving down in preparation for printing the copper slurry. Behind the copper slurry is the LTCC slurry that is deposited. Figure 3d shows the completion of copper slurry printing. The image on the right shows the mask for this print layer. This 3D printing process is repeated to allow printing using two different slurries.

### 3.2. Powder Size Measurement

The LTCC ceramic slurry and the copper slurry are ball milled to refine the powder size and to ensure that the refined powder is evenly distributed in the slurry. Figure 4 shows the SEM images of the LTCC and the size of the copper powder for as-received (a) and (b) and after ball milling (c) and (d). Particle analysis shows that the initial powder size for the LTCC and copper is 1.6~2.1 µm and 16.1~22.4 µm, respectively. After ball milling, the average refined powder size for the LTCC and copper is 1.459 μm and 15.3 μm, respectively. The shape of the refined particles is irregular, which increases the lamination and interlocking of slurry photopolymerization.

In terms of the average refined particle size for LTCC and copper, the diameter ratio of both is about 10 times. The average refined particle diameter for copper is 15.3 μm, which is greater than the value for LTCC. To ensure that the particles in the cured layer can be stacked, the target cured layer is 50 μm because this value is more than 3 times the average refined particle size for copper.

### 3.3. Exposure Time vs. Curing Depth

A powder-to-resin weight ratio of 80:20 results in poor fluidity; therefore, the slurry may not be deposited and cast on the tape, thus no exposure curing experiment is performed. This is consistent with the findings of Fernandes et al. study [16]. The maximum weight ratio of LTCC to resin to ensure good suspension in the slurry is 75 wt %. Figure 5 shows the relationship between the curing depth for slurries with different weight ratios for different exposure times. For the same exposure time, the greater powder content obtained the thinner cured specimen. When the cured thickness reaches the limit of thickness, increasing the exposure time does not increase the thickness of the cured layer. The LTCC particles are finer than those of copper so the stacking density of copper is less than that of LTCC. For the same powder content, photons penetrate deeper at low stacking densities so the cured thickness of the copper slurry is thicker than that of the LTCC slurry for the same exposure time.

Figure 5a shows that the LTCC slurry is cured to obtain a measurable layer thickness after exposure for 10 s. LTCC slurry with a weight ratio of 70:30 gives a cured thickness of 60 μm and 75 μm for exposure times of 10 and 20 s, respectively. The cured thickness for an exposure time of 10 s is greater than the target cured thickness so an exposure time of 20 s is the optimal time for curing the LTCC slurry to increase the curing bond strength between layers. 

Figure 5b shows the cured depth curve for the copper slurry. A copper slurry with a weight ratio of 70:30 gives respective cured layer thicknesses of 76, 121, and 137 μm for exposures of 5, 10, and 15 s. Thicker cured layers are produced using two other copper slurries. The greater powder content in the slurry produces a smaller volume shrinkage after sintering so the copper slurry with a weight ratio of copper powder to the resin of 70:30 is used to print the specimen and benchmark. The optimal exposure time for the copper slurry is 10 s because the target cured layer thickness is 50 μm. Table 1 shows the printing parameter for LTCC and Copper slurries. 

### 3.4. Thermogravical Analysis and Sintering Parameters

Figure 6 shows the relationship between heating temperature and weight loss for the TGA result. The resin begins to react when it is heated at 260 °C and there is significant weight loss between 350 and 500 °C. This shows that the heating rate for this temperature range cannot be fast to avoid generating large numbers of bubbles. The resin is completely vaporized at 500 °C, thus the heating rate is increased above 500 °C. The sintering temperature for LTCC and copper is about 850 °C, thus the maximum temperature for the sintering process is 850 °C.

This study uses a three-stage sintering process and the result of thermogravical analysis is shown in Figure 7. In the first stage, the green body is heated to 350 °C at a heating rate of 2 °C/min and maintained at that temperature for 1 h before the resin is evaporated. The temperature is then increased from 350 °C to 500 °C at a heating rate of 10 °C/min and then maintained at 500 °C for 1 h to ensure that all resin is evaporated. The temperature is then increased from 500 °C to 850 °C in 150 min and the specimen is sintered for 2 h at 850 °C, and then cooled in the furnace to room temperature at a rate of 50 °C/min. The success rate for sintering is increased by including glass in the composition of LTCC, but studies show that the composition of the glass also affects the dielectric loss [17].

### 3.5. Specimen Printing and Shrinkage Analysis

Figure 8 shows the printed LTCC and copper specimens that are used for shrinkage analysis. Figure 8a shows the printed LTCC specimen that is produced using a platform that ascends at 15mm/s. There is distortion at the corner, as indicated by the arrow. Figure 8b shows the LTCC green body that is printed using an ascension speed for the platform of 10 mm/s. This specimen features no distortion at the corner. Therefore, the speed at which the platform ascends is 10 mm/s. Copper specimens for shrinkage analysis can also be printed using the parameters shown in Figure 8c.

The dimensions of each printed specimen are measured and the specimens undergo sintering. Crack-free sintered parts are obtained using the proposed three-stage sintering parameters. Table 2 and Table 3 list the dimensions of each specimen for LTTC and copper and the average shrinkage ratio in each axial direction. The respective average shrinkage ratios for the sintered LTCC and copper specimens in the X, Y, and Z axes are 30.84%, 31.18%, 32.34% and 31.25%, 31.45, 31.9%. The experimental results show that the shrinkage ratios for these two slurries are very similar, which increases fusion when the two slurries are used to form a printed part and sintered simultaneously. 

The volume shrinkage for copper is 68.85% greater than that for LTCC ceramics (67.97%) because the particles in the copper powder are larger than those in the LTCC powder, thus there are more voids during free stacking. The resin occupies the stack voids after curing, but after sintering the resin vaporizes from the space so there is volume shrinkage. After sintering, the practical densities of LTCC and copper are 4.914 g/cm^3^ and 8.815 g/cm^3^, respectively. Both values are slightly lower than the theoretical density but within a 5% error margin.

### 3.6. Hardness and Flexural Strength

The measured hardness of the LTCC sintered part is between 533.6 HV and 540.6 HV. The indentation image for the Vickers hardness test is shown in Figure 9. Previous studies show that a hardness of 4.2~5.65 GPa is obtained using a pressure of 100MPa to make an LTCC device [15]. The LTCC is a mixture of CBS glass powder and stoichiometric cordierite ceramic powder that consists of various proportions of MgO, Al_2_O_3,_ and SiO_2_ (+99%). The hardness of the LTCC specimen that is sintered to 800 °C is 4.2 GPa. This value is greater than the average hardness for this study, but the LTCC composition is different from the composition for this study. This study also does not use a pressing process: powders are stacked freely, thus the density of sintered part is less than that of the pre-pressed sintered part.

The specification data for CCF given by the material supplier show that the flexural strength of pre-pressed sintered object is about 130~150 MPa. For this study, the three-point bending result for the LTCC green body and sintered part which produced using a 70:30 slurry is 89.3 MPa and 126.61 MPa, respectively. This value is about 11% less than the average value for a pre-pressed sintering object. This result is expected because no pre-pressing process is used for 3D printing in this study. 

Studies show that the glass composition in the LTCC slurry has an effect on flexural strength for sintering at a specific temperature [18]. The inclusion of Al_2_O_3_ as a filler in the LTCC material increases flexural strength [19] because Al_2_O_3_ exits the glass and forms tetrahedral (AlO_4_). This type of tetrahedral is more stable than other tetrahedral so the strength of the material increases [20]. Future studies will optimize the composition of LTCC powders for 3D printing applications.

### 3.7. Microstructural Observations

Figure 10 shows the microstructural SEM image for an LTCC sintered part, showing glass phase (grains), granular Al_2_O_3_ (indicated by arrowhead) and pores (indicated by circles). The presence of a glass phase indicates that the sintering temperature is correct, so crystallization occurs. However, there is still granular Al_2_O_3_, thus the temperature is not maintained for a sufficiently long period. Pores may also decrease the flexural strength to less than the samples that undergo pre-pressed sintering. Extending the holding time for the third stage from 120 min to 150 min inhibits the formation of granular Al_2_O_3_, but very tiny pores still exist.

The structure of the LTCC device is composed of ceramic and copper. To fabricate an LTCC device, the LTCC ceramic and copper must fuse without cracking. Figure 11a,b respectively show a printed benchmark green body and sintered parts. The sintering produces shrinkage so the sintered part is smaller than the green body, but there is no cracking on the surface. Figure 12a shows the SEM image of the interface microstructure of LTCC ceramic and copper. The interface of the ceramic and copper is fused and there are no pores. Figure 12b shows the embedded copper channel in the sintered LTCC device, which is electrically conductive, as verified by using a volt-ohm-milliammeter to measure the electrical resistance of 0.175 Ω. If the LTCC and copper are co-sintered, the oxidation of copper turns the LTCC from white to dark grey. After polishing, the copper regains its original color but the LTCC ceramics do not. This phenomenon may be avoided by using a vacuum sintering furnace.

## 4. Conclusions

This study proposes an innovative tape-casting multi-slurry 3D printing process used to produce a prototype machine. In terms of photocurable slurries, the powder content and printing parameters for LTCC and copper slurries are optimized to print the green body using two slurries. A three-stage sintering parameter is proposed using the TGA of slurry. The following summarizes the results of the experiment:Regarding the powder-to-resin weight ratio of LTCC and copper slurries, both can be deposited and cast on tape on demand and leveled by a scraper smoothly at 70:30. The respective exposure times to ensure a cured thickness of more than 50 μm are 20 and 10 s for the LTCC and copper slurries.The proposed three-stage sintering parameter allows the green bodies of LTCC and copper to be sintered without cracks or delaminations. The printed LTCC device, which contains a copper circuit, is fused after the sintering process because the shrinkage ratio in all directions for the two slurries is similar. The volume shrinkage for both slurries is also very comparable.When the green body of an LTCC device with a copper circuit is sintered, the copper is oxidized, thus the part changes color from white to dark gray. After polishing, the copper regains its original color. The sintered circuit has a resistance value of 0.175 Ω, demonstrating that it is electrically conductive.

## Figures and Tables

**Figure 1 materials-16-00585-f001:**
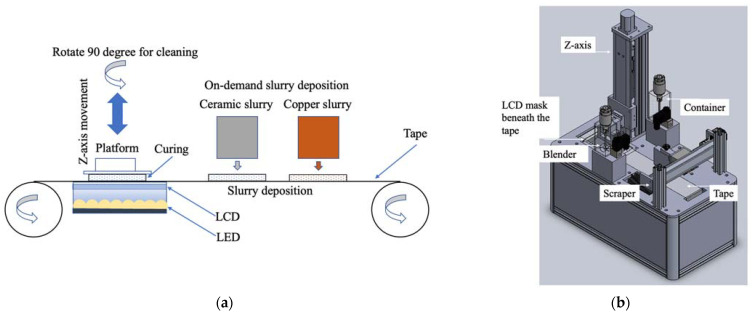
The proposed tape-casting multi-slurry 3D printing process: (**a**) the principle and (**b**) the proposed 3D printer.

**Figure 2 materials-16-00585-f002:**
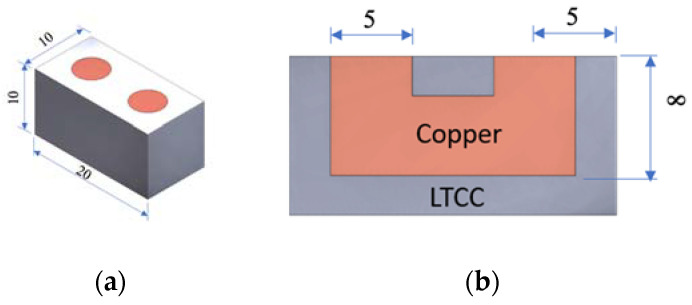
The proposed benchmark LTCC device: (**a**) dimensions and (**b**) cross-section.

**Figure 3 materials-16-00585-f003:**
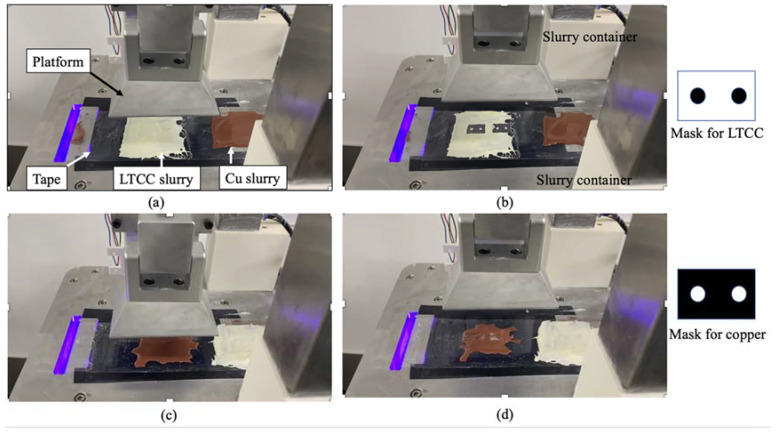
The tape-casting multi-slurry 3D printing process for the assembled printer. (**a**) depositing slurry on the tape; (**b**) photopolymerized LTCC adheres to the platform; (**c**) printing the Copper slurry; (**d**) the completion of copper slurry printing.

**Figure 4 materials-16-00585-f004:**
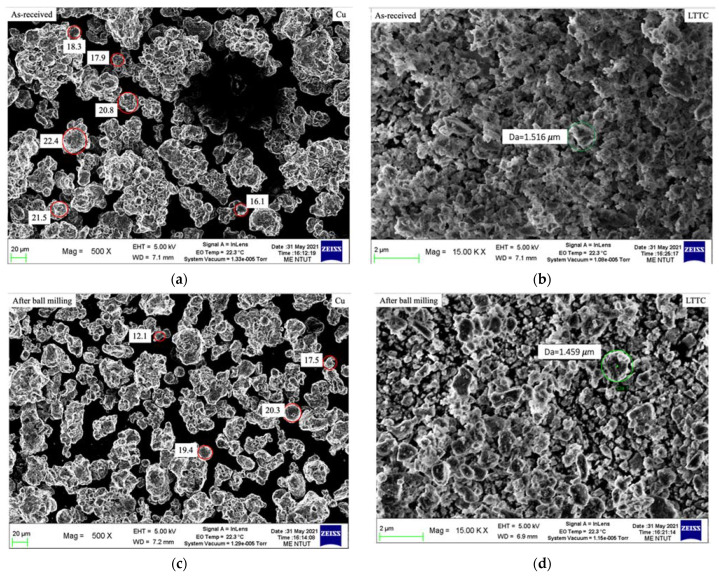
SEM images of LTCC and copper powder for as-received and after ball milling: (**a**) as-received of LTTC; (**b**) as-received of Cu; (**c**) after ball milling of LTTC; (**d**) after ball milling of Cu.

**Figure 5 materials-16-00585-f005:**
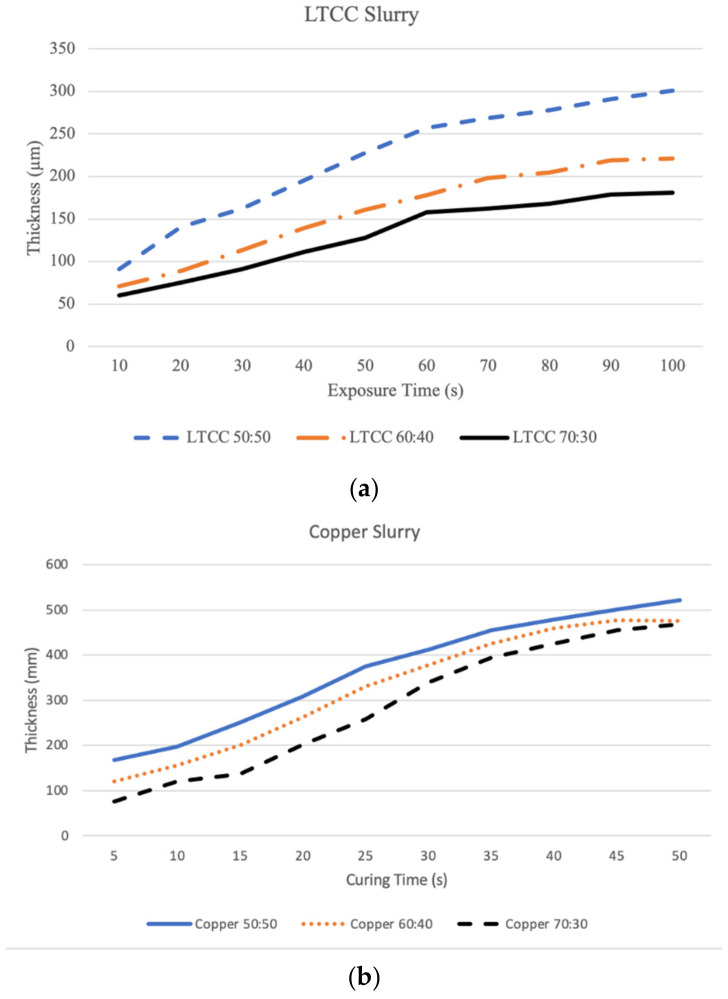
Cured layer thickness for different weight ratios and different exposure times: (**a**) LTCC and (**b**) copper slurry.

**Figure 6 materials-16-00585-f006:**
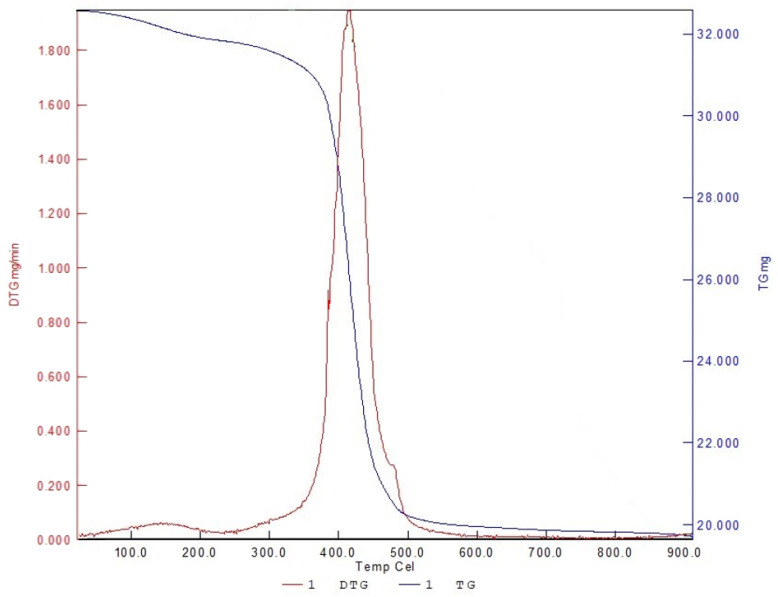
Result of thermogravical analysis.

**Figure 7 materials-16-00585-f007:**
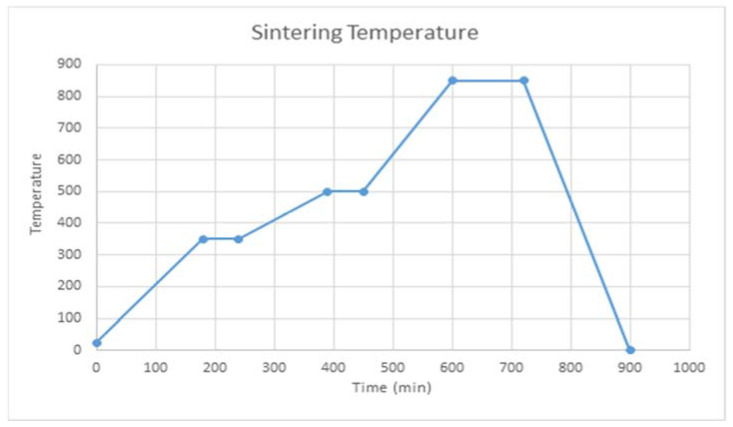
Three-stage sintering process.

**Figure 8 materials-16-00585-f008:**
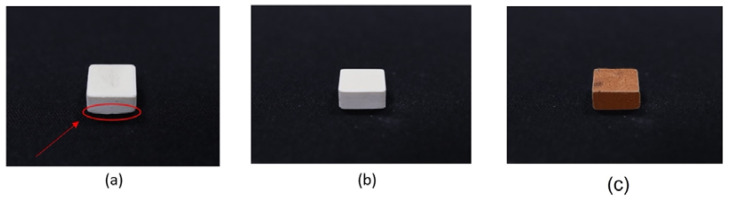
Printed specimens: (**a**) distortion at the corner of the LTCC green body, (**b**) successfully printed specimen with a 50 μm layer thickness for (**b**) LTCC and (**c**) copper slurries.

**Figure 9 materials-16-00585-f009:**
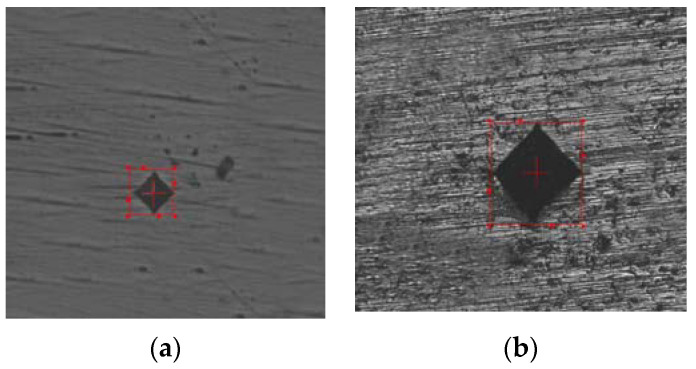
Indentation images of the sintered part: (**a**) LTCC and (**b**) copper.

**Figure 10 materials-16-00585-f010:**
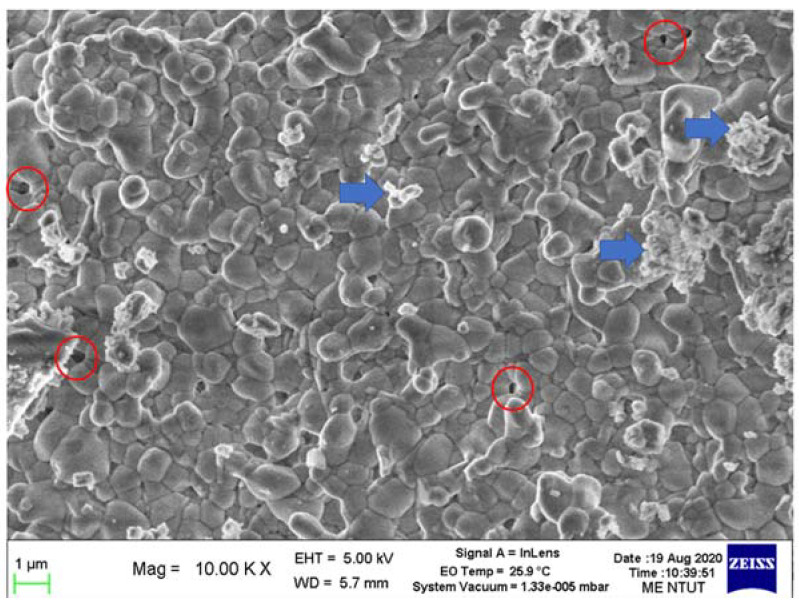
Microstructural SEM image of an LTCC sintered part.

**Figure 11 materials-16-00585-f011:**
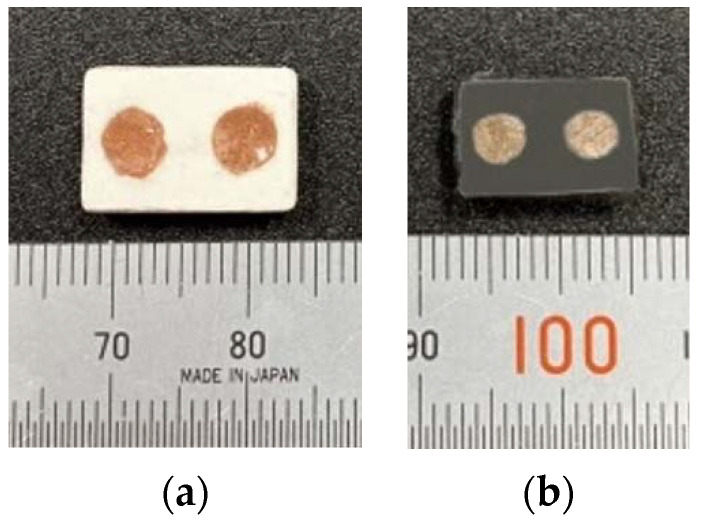
Printed benchmark for a green body (**a**) and a sintered part (**b**).

**Figure 12 materials-16-00585-f012:**
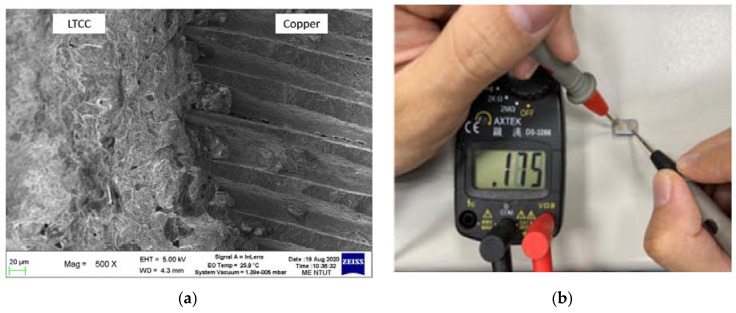
SEM image of the microstructure of the interface between the LTCC ceramic and copper (**a**) and the electrical conductivity test (**b**).

**Table 1 materials-16-00585-t001:** Printing parameters for LTCC and Copper slurries.

Parameter	LTCC Slurry	Copper Slurry
Average particle size	1.459 μm	15.3 μm
Exposure time (s)	20	10
Layer thickness (μm)	50	
Weight ratio of powder to resin	70:30	

**Table 2 materials-16-00585-t002:** Dimensions of the LTCC specimen before and after sintering.

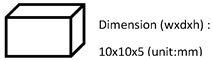		LTCC:Resin (70:30)					
		1	2	3	4	5	Average
Green Body	X (mm)	10.02	10.05	10.00	10.01	10.02	10.02
	Y (mm)	10.10	10.01	10.04	10.02	10.00	10.03
	Z (mm)	5.02	5.00	5.01	5.02	5.03	5.02
	Density (g/cm^3^)	2.413	2.371	2.466	2.432	2.544	2.45
	Volume (cm^3^)	0.512	0.516	0.515	0.525	0.536	0.52
	Weight (g)	1.236	1.224	1.269	1.276	1.364	1.274
Sintering Part	X (mm)	6.98	6.91	6.90	6.91	6.95	6.930
	Y (mm)	7.07	6.84	6.85	6.86	6.91	6.906
	Z (mm)	3.39	3.40	3.41	3.39	3.38	3.394
	Density (g/cm^3^)	4.927	4.834	5.050	4.813	4.947	4.914
	Volume (cm^3^)	0.165	0.175	0.159	0.166	0.169	0.167
	Weight (g)	0.813	0.846	0.803	0.799	0.836	0.819
Shrinkage %	X (%)	30.34%	31.24%	31.00%	30.97%	30.64%	30.84%
	Y (%)	30.00%	31.67%	31.77%	31.54%	30.90%	31.18%
	Z (%)	32.47%	32.00%	31.94%	32.47%	32.80%	32.34%
Shrinkage (vol % )		67.79%	66.10%	69.11%	68.36%	68.48%	67.97%
Weight Loss (wt %)		34.22%	30.88%	36.72%	37.38%	38.71%	35.58%

**Table 3 materials-16-00585-t003:** Dimensions of the Copper specimen before and after sintering.

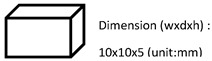		Cu Powder:Resin (70:30)					
		1	2	3	4	5	Average
Green Body	X (mm)	10.01	10.08	10.11	10.01	10.03	10.05
	Y (mm)	10.08	10.03	10.07	10.02	10.00	10.04
	Z (mm)	4.99	5.04	5.08	5.01	5.01	5.03
	Density (g/cm^3^)	4.812	4.817	4.821	4.812	4.814	4.82
	Volume (cm^3^)	0.503	0.510	0.517	0.503	0.503	0.51
	Weight (g)	2.423	2.455	2.493	2.418	2.419	2.442
Sintering Part	X (mm)	6.93	6.90	6.91	6.92	6.88	6.908
	Y (mm)	6.87	6.88	6.91	6.86	6.89	6.882
	Z (mm)	3.33	3.32	3.36	3.30	3.30	3.322
	Density (g/cm^3^)	8.812	8.818	8.815	8.813	8.817	8.815
	Volume (cm^3^)	0.1585	0.1576	0.1604	0.1567	0.1564	0.1579
	Weight (g)	1.397	1.390	1.414	1.381	1.379	1.392
Shrinkage %	X (%)	30.77%	31.55%	31.65%	30.87%	31.41%	31.25%
	Y (%)	31.85%	31.41%	31.38%	31.54%	31.10%	31.45%
	Z (%)	33.27%	34.13%	33.86%	34.13%	34.13%	33.90%
Shrinkage (vol % )		68.51%	69.07%	68.98%	68.83%	68.87%	68.85%
Weight Loss (wt %)		42.34%	43.38%	43.28%	42.90%	42.98%	42.98%

## Data Availability

Not applicable.

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
