# Peer review of "Development of a Novel Tape-Casting Multi-Slurry 3D Printing Technology to Fabricate the Ceramic/Metal Part"

_materials, 2023, doi:10.3390/ma16020585_

Round 1
Reviewer 1 Report
The manuscript should be published after minor revision.
Lines 114-115. The wavelength of LED should be mentioned.
Lines 348-349. LTCC flexural strength is given only for sitered part. However, it is worth to mention flexural strength of greenbody state of LTCC and flexural strength of solely printed copper slurry and LTCC.
Reviewer 2 Report
Overall this paper has the potential to be published in this journal. However, the followings should be addressed before publication:
(1) There are many typos and errors in the text that makes it hard to understand the concept. Please proof read the paper and fix the mistakes.
(2) this is a custom-made 3D printer. The authors need to address how this study can be useful for the broader range of audiences who do not have access to this laboratory-developed 3D printer?
(3) Can you perform the same experiments with parts 3D printed using commercial 3D printers?
Therefore, the paper is Accepted with Major Revision.
Reviewer 3 Report
The fonts in the plots are not visible. Please increase the size. There is no description of the machine. What is the expected cost of the parts to be manufactured by the new printer? Is it economical? What kind of software arrangements are needed to run the machine? How the slicing of the object taken place while fabricating it? 3D printing helps achieve material complexity, functional complexity, and shape complexity. However, the part authors reported is very simple. How can we justify that the new arrangments fulfill those?
Round 2
Reviewer 2 Report
There are still many grammar and typos in the submitted manuscript and I cannot recommend this paper for publication unless they are addressed. The mistakes are causing misunderstanding of the concept shown in the paper.
To mention only few of them, please refer to lines: 12, 35, or 114. These are only few obvious grammar and typo mistakes.
Please address the grammar and typos.
Additionally, authors need to mention in the text how this technique can be used by others if needed. Step by step instruction needs to be explained to make this study replicable.
Author Response
First of all, I would like to express my sincere appreciation for the constructive comments, considerable time and significant efforts on my manuscript. Regarding the proposed modifications and points to be addressed, response is as follows:
1. To mention only few of them, please refer to lines: 12, 35, or 114. These are only few obvious grammar and typo mistakes.
Response:
Thanks for the constructive comment and suggestion, the author considers the readability of this manuscript very seriously, and already made some changes and grammar checks.
In line 12 we made some changes also grammar checks:
Printing ceramic/metal parts increase the number of applications in additive manufacturing technology, but printing different materials on the same object with different mechanical properties will increase the difficulty of printing.
In Line 35:
Additive Manufacturing (AM), which is also known as three-dimensional (3D) printing, is a method to fabricate physical parts from a virtual design model
In Line 114:
The light emitting diode (LED) module has a 10Watt output power with 405 nm wavelength
The other grammar and typo check also already been revised by the author.
2. Authors need to mention in the text how this technique can be used by others if needed. Step by step instruction needs to be explained to make this study replicable.
Response:
Thanks for the constructive comment. In the section Principle of tape-casting multi-slurry 3D printing technology, we made the methods clear, and how the proposed printing works step by step (Lines 109-126), section 3-1 shows the printing process. Since the author applied for the patent and still carrying out the technical transfer to the company, the detail of the machine can’t be shown up. However, the author offers the contact information for readers to further contact if they need any help
We used the tracking mode in this revision manuscript so that the reviewer can easier to check our modification. Thanks for reviewer comment to improve this study can be more scientific present the contribution in further industry application.
Best regards
Sincerely
The authors respect.